# Feature Overlapping: The Computational Redundancy Caused by Repeated Features Across Different Time Steps in SNNs

## Abstract

Spiking neural networks (SNNs) have the potential advantage of building large-scale energy-efficient network. However, the high training cost caused by multiple time steps currently limits the application of SNNs. To address this, we break away from the traditional approach of reducing the number of time steps and investigate feature redundancy between time steps. By jointly unfolding the computational process of SNNs across both temporal and spatial dimensions, we are the first to discover the Feature Overlapping Phenomenon, providing new insights for improving SNNs training paradigms. Our Temporal Differential Decoupling (TDD) method successfully separates dynamic and static features, reducing redundant computations. By transforming the feature space into the differential domain, it addresses the issue of the original computational domain's inability to effectively filter sensitive information. In the differential domain, we propose the Gradient Sensitivity Criterion (GSC), which helps further reduce training costs and avoids the loss of important feature information. This paper introduces the Differential Domain Low-Sparsity Approximation (DDLA) algorithm, which significantly reduces computational resource consumption while maintaining computational accuracy by adjusting the filtering ratio. Experimental results show that we achieved up to an 80.9% reduction in the number of spikes per timestep and a total spike count reduction of up to 57.8%, significantly reduce the inference cost of SNNs.

## 1 Introduction

Spiking Neural Networks (SNNs) process all spiking events to process and represent information, offering advantages such as low power consumption and high biological fidelity compared to Artificial Neural Networks (ANNs) Allen et al. (2009); Guo et al. (2017). As a third-generation neural network model, they have garnered widespread attention Kasabov & Capecci (2015); Zhang et al. (2013). However, existing SNN networks rely on rate coding to encode features into spike sequences, compensating for quantization errors through joint expectations over multiple time steps Yang et al. (2021). This results in computational cost that increases exponentially with the number of time steps, posing significant challenges for inference and training in large-scale networks, which hinders the application of SNNs.

Currently, to address the issue of high computational cost over time steps, existing work primarily focus on reducing the number of time steps. Li et al. Li et al. (2023) proposed SEENN-I and SEENN-II, which explore fine-grained adjustments to the number of time steps during the inference phase. Yao et al. Kim et al. (2023) studied the dynamics of temporal information, redistributing time information to achieve delay-adaptive decision-making. However, these works mainly focus on reducing the number of time steps through dynamic configuration Shen et al. (2024). Given that the current SNNs already operate within five time steps, further reducing time steps results in a significant loss of accuracy. Therefore, we believe that to address the high computational cost of large-scale SNNs, the solution should not be to directly reduce time steps but rather to focus on whether there is redundancy between time steps.

Through experiments, we found that between consecutive time steps, due to weight sharing and identical image encoding, the firing patterns of neurons are often very similar. This led us to the following thought:

- Are there redundant features in the multiple time steps of an SNN?

- How can we decouple effective features and redundant features?

- How can we reduce the computation associated with redundant features?

We present a frst-of-its-kind study to discover that certain feature information in the same time sequence repeatedly appears across time steps, a phenomenon we define as "feature shadowing". Previous works focus only on feature encoding and information representation capabilities. Due to the inability to decouple redundant features, they believe that indiscriminate parallel computation across multiple time steps is necessary for rate coding. Through clever Temporal Differential Decoupling (TDD), we discover feature repetition and achieve redundancy separation. This provides new possibilities for handling challenge of high training and inference costs in large-scale SNNs.

We propose the TDD method, which effectively decouples dynamic and static features. The features vary with the time steps, are defined as dynamic features, representing the information gain for the current time step. Transforming the feature space into the differential domain, making it possible to filter out key features. In the original computational domain, crucial dynamic information could be overlooked due to interference from static features. However, in the differential domain, sensitive features are clearly identified, which avoids the negative impact of irrelevant features on result accuracy. We propose the Gradient Sensitivity Criterion (GSC) to filtering Sensitivity Features. Based on TDD and GSC, we propose Differential Domain Low-sparsity Approximation (DDLA) algorithm. By adjusting the filtering ratio, our algorithm is possible to achieve any desired level of computational precision while dramatically lowering the consumption of computational resources.

Our contributions are summarized as follows:

1. We are the first to discover the Feature Overlapping Phenomenon, which not only reveals the issue of computational redundancy in SNN training paradigms but also provides new insights for improving computational efficiency.

2. Temporal Differential Decoupling method effectively decouples dynamic and static features by transforming the feature space into the differential domain, which allows for better filtering of key features. We propose Differential Domain Low-sparsity Approximation algorithm, which significantly reduces computational resource consumption while ensuring computational precision.

3. We validated the effectiveness of our approach through comprehensive testing on both static and event-based datasets across various network architectures. While ensuring accuracy comparable to baseline models, we achieved up to an 80.9% reduction in the number of spikes per timestep and a total spike count reduction of up to 57.8%, significantly enhancing computational efficiency.

## 2 RELATED WORK

### 2.1 ANN-TO-SNN CONVERSION

The ANN-to-SNN conversion is a widely used method for obtaining SNNs by leveraging pre-trained ANNs Ho & Chang (2021); Wang et al. (2024). This approach converts an ANN with ReLU activation functions into an SNN with spike-based activations, typically involving scaling the ANN's weights and biases to ensure compatibility with the spiking characteristics of SNNs Wang et al. (2022). Conversion methods often include scaling operations, such as weight normalization and threshold balancing, to better align the dynamic range of activations in SNNs with those in ANNs Hwang & Kung (2024); Kundu et al. (2021). However, these conversion methods are challenged by significant conversion errors and inference delays, which make it difficult to effectively handle highly dynamic or deep neural network architectures.

## 2.2 DIRECT TRAINING

Direct training of SNNs, without relying on pre-trained ANNs, has gained considerable attention in recent years Lee et al. (2016); Yin et al. (2021), particularly for its advancements in reducing time steps Bohte (2011), improving computational efficiency Sengupta et al. (2019), and enhancing network performance Zheng et al. (2021). This approach typically involves using surrogate gradient methods to overcome the challenges of gradient computation due to the discontinuity and non-differentiability of spiking neurons Fang et al. (2021). Recent developments have made it possible to achieve efficient training with fewer time steps by optimizing surrogate gradient, firing thresholds and loss functions Fan et al. (2023). However, despite the reduction in time steps, the computational cost per time step remains high. Reducing the computational cost is a critical focus for future research.

## 2.3 DYNAMIC TIME STEP

The computational cost of SNNs increases exponentially due to their multiple steps processing of spiking events Dominguez-Morales et al. (2018). To address this issue, current research has primarily focused on two approaches: dynamically adjusting the number of time steps Li et al. (2023) and dynamically redistributing temporal information Kim et al. (2023); Shen et al. (2024). Methods that adjust the number of time steps aim to dynamically modify the time steps of SNNs based on the characteristics of the input samples Meng et al. (2022). For example, SEENN-I uses confidence scores, and SEENN-II employs a policy network to determine the inference time steps. Another set of methods use Bayesian learning through deep ensembles or variational inference to aggregate information across time steps. However, these approaches primarily focus on configuring time steps and offer limited reductions in computational costs. We believe that, with the number of time steps recently reduced to fewer than five, the focus should shift to targeting redundancy in computations across time steps.

## 3 PRELIMINARY

In SNN, the membrane potential of Integrate-and-Fire(IF) neurons gradually changes based on the input current it receives and the leaky characteristics of the membrane potential. The pre-activation membrane potential of a neuron can be described as:

$$\tilde{u}_l(t) = u_l(t-1) + W_l s_{l-1}(t) \tag{1}$$

where $\tilde{u}_l(t)$ denotes the pre-activation membrane potential of the neuron in $l$ layer at time $t$, $u_l(t-1)$ denotes the membrane potential of the neuron at the previous time step $t-1$, $W_l$ denotes the synaptic weight matrix between neurons in layer $l-1$ and layer $l$, $s_{l-1}(t)$ denotes the spike from neurons in layer at $l-1$ time $t$, which is a binary value representing whether the neurons fired a spike (1) or not (0).

When the membrane potential exceeds a certain threshold, the neuron fires a spike:

$$s_l(t) = Hea\left(\tilde{u}_l(t) - V_{th}\right) \tag{2}$$

where $s_l(t)$ denotes the spike output of the neuron in layer $l$ at time $t$, which is a binary value. It equals 1 if the neuron fires a spike and 0 if it does not. $Hea(x)$ represents the Heaviside step function, determining whether the neuron fires a spike based on the difference between the membrane potential and the threshold. $V_{th}$ represents the firing threshold of the neuron.

The residual potential remaining after a neuron fires a spike can be described as:

$$u_l(t) = \tilde{u}_l(t)\left(1 - s_l(t)\right) = \tilde{u}_l(t) - r_l(t)s_l(t) \tag{3}$$

$$r_l(t) = \begin{cases} u_l(t) & s_l(t) = 1 \\ 0 & s_l(t) = 0 \end{cases} \tag{4}$$

where $u_l(t)$ denotes the residual potential of the neuron in layer $l$ at time $t$, $r_l(t)$ denotes the reset potential of the neuron in layer $l$.

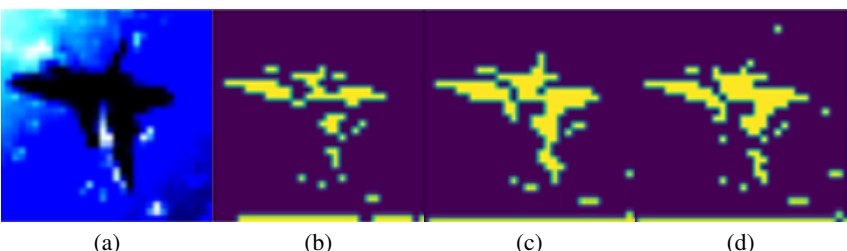

|     |     |     |     |
| :-: | :-: | :-: | :-: |
| (a) | (b) | (c) | (d) |

Figure 1: Feature maps of SNN convolutional layers at different time steps. Figure (a) shows the original image from the CIFAR-10 dataset, while Figures (b), (c), and (d) depict the feature maps output by the first convolutional spiking layer at time steps $t = 1$, $t = 2$, and $t = 3$ during image classification using ResNet-19 SNN. It can be observed that the feature maps at different time steps are highly similar.

## 4 FEATURE OVERLAPPING PHENOMENON

Through experiments, we found that between adjacent time steps, due to weight sharing and similar image encoding, the firing patterns of neurons are often very similar, as shown in figure 1. Given this similarity, we hypothesize that there may be overlapping and redundant feature information across different time steps.

In this section, we expand and derive the SNN computation process from both spatial and temporal dimensions. Temporal Differential Decoupling (TDD) separate dynamic information from static information. This leds to the discovery of the Feature Overlapping Phenomenon, which proves that the current SNN training paradigm indeed contains computational redundancy.

### 4.1 TEMPORAL DIFFERENTIAL DECOUPLING

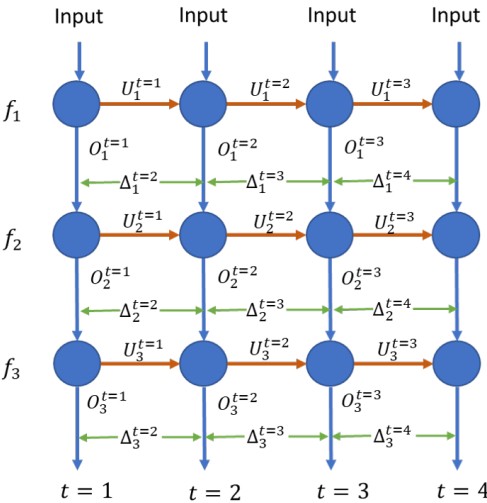

Figure 2: Illustration of the time-step unfolding process in SNN computation. This diagram demonstrates the unfolding process of SNNs computation across time steps. The blue circles represent neurons, with the first column depicting neurons with a IF mechanism. The orange arrows indicate the transmission of signals between layers, while the green dashed lines show the flow of signals across time steps.

From the spatial dimension, the core feature of information transmission between layers in SNN is that each neuron's output depends on the output of the previous layer and the accumulated residual

potential input from previous time steps. The output can be expressed as:

$$O_l^T = F\left(f_l\left(O_{l-1}^T\right) + U_l^{T-1}\right) \tag{5}$$

where $O_l^t$ denotes the output of the neuron in layer $l$ at time step $t$, $U_l^{t-1}$ denotes residual potential from the previous time step $t-1$, $f_l$ denotes the linear transformation applied to the output of the previous layer, $F$ is the nonlinear activation function.

From the temporal dimension, to capture the information about changes between time steps, the output features at each time step are split into the overlapping part and the differential part. The overlapping features, which do not change between two consecutive time steps, are defined as Static Feature. The differential features represent those that change with the time steps and contribute to the information gain at each time step, which we define as Dynamic Feature. The output can be expressed as:

$$O_l^T = \underbrace{O_l^{T-1}}_{Static\ Feature} + \underbrace{\Delta_l^T}_{Dynamic\ Feature} = \sum_{t=1}^{T} \Delta_l^t \tag{6}$$

where $\Delta_l^t$ donates the differiential part of the later time step relative to the previous time step.

Taking both the temporal and spatial dimensions into account, the neuron in layer $l$ at time step $t$ is influenced by all the historical changes $\Delta_{l-1}^T$ passed from layer $l-1$. Since $f_l$ is a linear function, output can be expressed as:

$$O_l^T = F\left(f_l\left(O_{l-1}^T\right) + U_l^{T-1}\right) = F\left(f_l\left(O_{l-1}^1\right) + f_l\left(\Delta_{l-1}^2\right) + f_l\left(\Delta_{l-1}^3\right) + \cdots + f_l\left(\Delta_{l-1}^T\right) + U_l^{t-1}\right)$$

$$= F\left(\sum_{t=1}^{T-1} f_l\left(\Delta_{l-1}^t\right) + f_l\left(\Delta_{l-1}^t\right) + U_l^{t-1}\right) \tag{7}$$

Here is an example of the computation process for $l = 2$ neurons of an SNN with $t = 4$, with and without TDD, as shown in Table 1. The first equals sign represents the original SNN calculation method, and the second equals sign shows the computation method after applying TDD.

Table 1: Time-step and feature component analysis in SNN computation process.

| Output | SNN computation model | Feature overlapping part | Feature differential part | Residual potential |
|---|---|---|---|---|
| $O_2^1$ | $= F\left(f_2\left(O_1^1\right)\right)$ | $= F($ | $+f_2\left(\Delta_1^1\right)$ | $)$ |
| $O_2^2$ | $= F\left(f_2\left(O_1^2\right) + U_2^1\right)$ | $= F\left(f_2\left(\Delta_1^1\right)\right.$ | $+f_2\left(\Delta_1^2\right)$ | $+U_2^1)$ |
| $O_2^3$ | $= F\left(f_2\left(O_1^3\right) + U_2^2\right)$ | $= F\left(f_2\left(\Delta_1^1\right) + f_2\left(\Delta_1^2\right)\right.$ | $+f_2\left(\Delta_1^3\right)$ | $+U_2^2)$ |
| $O_2^4$ | $= F\left(f_2\left(O_1^4\right) + U_2^3\right)$ | $= F\left(f_2\left(\Delta_1^1\right) + f_2\left(\Delta_1^2\right) + f_2\left(\Delta_1^3\right)\right.$ | $+f_2\left(\Delta_1^4\right)$ | $+U_2^3)$ |
| $O_l^t$ | $= F\left(f_2\left(O_{l-1}^t\right) + U_l^{t-1}\right)$ | $= F\left(f_2\left(\Delta_{l-1}^1\right) + f_2\left(\Delta_{l-1}^2\right) + f_2\left(\Delta_{l-1}^3\right) + \cdots\right.$ | $+f_2\left(\Delta_{l-1}^t\right)$ | $+U_l^{t-1})$ |

## 4.2 DEFINITION OF FEATURE OVERLAPPING PHENOMENON

In the original SNN model, a full computation is performed at each time step to extract features specific to that moment (i.e., the orange section of the table 1). Although each computation extracts the feature information for that particular time step, due to weight sharing and the same input image, there is significant overlap in the features extracted across different time steps. Performing an XOR operation on the feature maps of two adjacent time steps defines the feature differential part (i.e., the red section of the table 1), representing the change in features at the next time step relative to the shifted features. This portion is the feature information that needs to be computed for the current time step. Temporal feature components are often repeatedly calculated across multiple time steps (i.e., the blue section of the table 1). These repeated computations do not introduce new information but rather process the same information multiple times. We define this phenomenon as follows 1.

**Definition 1.** *Let $O_l^t$ represent the output feature map of layer $l$ at time step $t$. Assume that this feature map is generated from the input $O_l^{t-1}$ and parameters $W_l$ through the function $f$: $O_l^t = f\left(O_l^{t-1}; W_l\right)$. we define the feature difference between time steps $t$ and $t-1$ as $\Delta_l^t = O_l^t \oplus O_l^{t-1}$,*

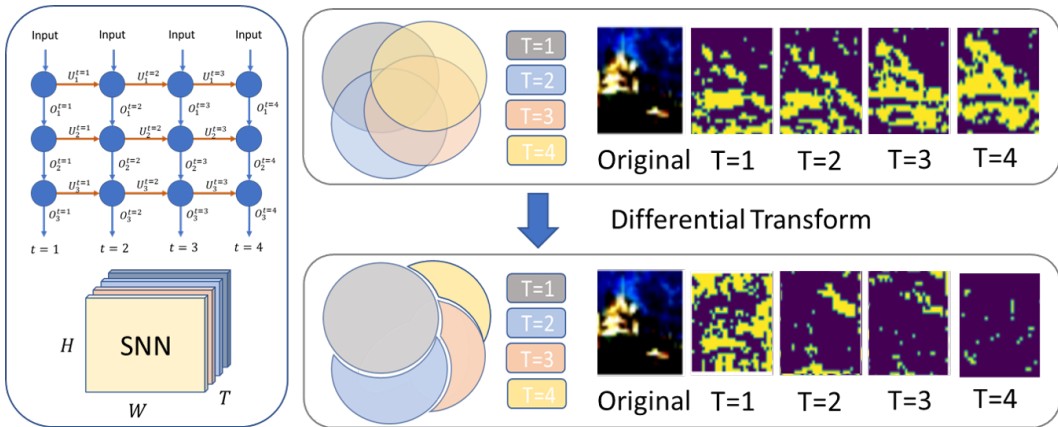

Figure 3: Illustration of of the Differential Transformation in SNN. The left side shows the structure of the SNN model and the temporal propagation of the input signals. The top right illustrates the original feature sequence across different time steps (T=1 to T=4), where noticeable feature overlapping occurs. The bottom right presents the image sequence after differential transformation, which removes redundant features between time steps.

*where $\oplus$ represents XOR operation.* **Feature Overlapping** *occurs when, for certain $l$ and $t$, the norm of the difference $\Delta_l^t$ is smaller than a certain threshold $\epsilon$, i.e., $\|\Delta_l^t\| < \epsilon$, which indicates that there is a significant overlap between the features at $t$ and $t-1$.*

This phenomenon refers to the overlap or redundancy between neuron activities or features across different time steps. For static features, the results from the previous time step can be directly reused, thereby reducing the computation cost. Consequently, after transforming into the differential domain, the more zero elements in $\Delta_l^t$, the fewer dynamic features, and the more pronounced the Feature Overlapping Phenomenon becomes, leading to a more effective reduction of redundant computation through feature reuse.

The discovery of this new phenomenon reveals computational redundancy in the current SNN training paradigm, offering new insights for improving computational efficiency, adaptive computation, and refining the training paradigm.

## 5 DIFFERENTIAL DOMAIN LOW-SPARSITY APPROXIMATION ALGORITHM

In the differential domain, zero elements have a clear meaning: they indicate no change between adjacent time steps. After TDD, dynamic features focus on key changes rather than the absolute values of the original features, avoiding ignoring crucial information about dynamic changes due to interference from static features. Therefore, it is possible to directly assess which features have a significant impact on the output, called Sensitive Features. Through sensitivity analysis, irrelevant or less impactful features can be set to zero, further reducing the computational cost. This is especially beneficial in resource-constrained systems, where it can significantly reduce power consumption and latency.

### 5.1 GRADIENT SENSITIVITY CRITERION

Here, we propose the Gradient Sensitivity Criterion (GSC) for selecting sensitive features. Assuming the selection of the top $k$ sensitive features $\Delta_l'^t = k\Delta_l^t$, it is necessary to minimize the error of the loss function:

$$\left|\Delta\mathcal{L}\left(O_l^t\right)\right| = \left|\mathcal{L}\left(O_l^t \mid W\right) - \mathcal{L}\left(O_l'^t \mid W\right)\right| \tag{8}$$

where $O_l'^t$ represents the approximate value of the output, $\mathcal{L}(O_l^t)$ represents the loss function value before sensitivity analysis and $\mathcal{L}(O_l'^t \mid W)$ represents the loss function value after c certain features to zero. The independence assumption during training helps in simplifying the optimization process,

even though the true behavior of the network is more complex. Approximating $\mathcal{L}(O'^t_l \mid W)$ with a first-order Taylor polynomial near $O^t_{l,i} = 0$ based on the Taylor expansion method, we have:

$$\mathcal{L}\left(O'^t_l \mid W\right) = \mathcal{L}\left(O^t_l \mid W\right) - \frac{\delta\mathcal{L}}{\delta O^t_{l,i}} O^t_{l,i} + R_1\left(O^t_{l,i} = 0\right) \tag{9}$$

where $R_1\left(O^t_{l,i} = 0\right)$ is the first order remainder. By substituting Eq. 9 into Eq. 8 and ignoring the remainder, we have $\zeta\left(O^t_{l,i}\right) : \mathbb{R}^{H \times W \times T} \to \mathbb{R}^+$:

$$\zeta\left(O^t_{l,i}\right) = \left|\Delta\mathcal{L}\left(O^t_l\right)\right| = \left|\mathcal{L}\left(O^t_l \mid W\right) - \frac{\delta\mathcal{L}}{\delta O^t_{l,i}} O^t_{l,i} - \mathcal{L}\left(O^t_l \mid W\right)\right| = \left|\frac{\delta\mathcal{L}}{\delta O^t_{l,i}} O^t_{l,i}\right| \tag{10}$$

Finally, we propose the Gradient Sensitivity Criterion as follows.

**Definition 2.** *Let the model's output at time $t$ be $O^t_{l,i}$, and the corresponding loss function be $\mathcal{L}\left(O^t_l \mid W\right)$, where $W$ are the parameters of the model. To measure the impact of an output feature $O^t_{l,i}$ on the loss function $\mathcal{L}\left(O^t_l \mid W\right)$, define the **Gradient Sensitivity Criterion** as: $\zeta\left(O^t_{l,i}\right) = \left|\frac{\delta\mathcal{L}}{\delta O^t_{l,i}} O^t_{l,i}\right|$*

When a neuron's gradient is large, it indicates that it has a stronger response to the task as a feature detector and should be retained during inference. This criterion requires accumulating the gradients of the features fed into the neuron at that moment, gradually selecting the features that contribute most to the task, thereby optimizing network computation.

## 5.2 OVERALL ALGORITHM

We propose a novel Differential Domain Low Sparsity Approximation algorithm (DDLA) based on TDD , complemented by GSC. The process begins with the transformation of feature maps from SNNs into the differential domain, where the focus is primarily on the differential increments of information. These increments are crucial as they represent the additional information that can influence subsequent layers. Our method specifically targets sensitive neurons that have a significant impact on the outputs of subsequent layers, while ignoring those with minimal influence. The rationale behind this approach is that if the output of a layer is insensitive to the next layer's response, it implies that any output from the current layer does not affect the subsequent layer, allowing us to set the output of insensitive neurons to zero to reduce computational load. Additionally, the discrepancy between the actual and expected outputs at each layer is incorporated into the overall loss function for optimization. Through iterative refinement, the algorithm ensures that the output from each layer significantly affects the following layer, thereby enhancing the overall efficiency and performance of the network. The detailed algorithm 1 is shown in the appendix.

## 6 TRAINING APPROXIMATION ERROR

Ignoring feature dependencies, the process of passing dynamic features while reusing static features and the process of setting features to zero after sensitivity analysis are sources of accuracy loss. Based on the error analysis formula of Taylor's theorem, we can derive the following theorem 1. The detailed derivation can be found in the appendix.

**Theorem 1.** *Given an L-layered SNN intended to inference $T$ time-steps, For layer $l$, according to Feature Overlapping and Taylor expansion to approximate the nonlinear activation function $F$, it satisfies:*

$$O^t_l \approx F\left(f_l\left(O^{t-1}_{l-1}\right) + U^{t-1}_l\right) + \nabla\hat{f}^\top_s\left(f_l\left(\Delta'^t_{l-1}\right)\right) f_l\left(\Delta'^t_{l-1}\right) \tag{11}$$

*where $\hat{f}_s(f_(O^{t-1}_{l-1}) + U^{t-1}_l)$ is the overlapping feature. This part reuses the output from the previous time step without recalculating it. $f_l(\Delta'^t_{l-1})$ represents the function operation on the feature changes at the current time step after sensitivity analysis, which is much sparser compared to the original feature map. $\nabla\hat{f}^\top_s(X)$ is the first-order derivative matrix of the spiking activation function.*

*If each component of $f_l(\Delta'^t_{l-1})$ is independent, let $\Gamma_l = f_l(O^{t-1}_{l-1}) + U^{t-1}_l = [\gamma_1, \gamma_2, \cdots, \gamma_n]$, the error $\epsilon^*$ satisfies*

$$\epsilon^* = \sum_{l=1}^{L} \sum_{i,j} \left( \frac{\partial \hat{f}_s(\Gamma_l)}{\partial \gamma_i} \cdot \frac{\partial \hat{f}_s(\Gamma_l)}{\partial \gamma_j} \right) \mathrm{Cov} \left( f_k \left( \Delta'^t_{l-1} \right)_i, f_k \left( \Delta'^t_{l-1} \right)_j \right)^2 \tag{12}$$

Theorem 1 explains how approximation errors arise due to decoupling dynamic features and static features. This is helpful for designing more efficient and accurate SNN models, where controlling these covariance terms can reduce errors.

In order to limit the algorithm error, we aim to obtain the expression of the supremum on $\|\epsilon^*\|$, and analyze the factors that influence the training approximation error. It is based on the following assumption:

**Assumption 1.** *The lower bound of the membrane potential $u_l(t) \in \mathbb{R}^{N \times T}$ is determined by ($\chi = \{0, 1\}$) : $\varphi = \sup_{s_l \neq 0, s_l \in \chi^{N_l-1}} \|u_l(t)\|_\infty \geq 0$*

The error is closely related to the neuron firing rate. From Assumption 1, we can obtain the upper and lower bounds of the membrane potential norm, which allows us to derive the range of the neuron firing rate, and thus determine the upper bound of the error, as shown in Theorem 2. The detailed derivation can be found in the appendix.

**Theorem 2.** *Given an $L$-layered SNN intended to inference $T$ time-steps, For layer $l$ with $N$ neurons, the membrane potential $u_l(t) \in \mathbb{R}^{N \times T}$ satisfies*

$$\varphi \leq \|u_l(t)\|_2 = \left\| W_l h_{l-1}(t) - \sum_{i=1}^{t} r_l(i) s_l(i) \right\|_2 \leq \sqrt{mn} V_{th} \tag{13}$$

*where $h_l(t) = \sum_{i=1}^{t} s_l(i)$ is the firing rate. It satisfies*

$$\frac{W_l h_{l-1}(t) - \sqrt{mn} V_{th}}{t V_{th}} \leq h_l(t) \leq \frac{W_l h_{l-1}(t) - \varphi}{t V_{th}} \tag{14}$$

*Therefore, the training error satisfies*

$$\|\epsilon^*\|_2 \leq \sum_{l=1}^{L} \frac{4 \left( \sqrt{N} V_{th} - \varphi \right)^2}{k^2 N T^2 V_{th}^2} \|W_l\|_2^2 \left\| \nabla f_s^T(\Gamma_l) \right\|_2^2 \tag{15}$$

*where $V_{th}$ is the threshold potential, $k$ is the filtering ratio of sensitivity analysis.*

It is worth noting that increasing the number of time steps can significantly reduce algorithmic errors. The more time steps there are, the greater the redundancy in the temporal feature maps, and the more effective our method becomes in reducing computational complexity. At the same time, the joint distribution of multiple time steps also compensates for algorithmic errors. Additionally, the filtering ratio $k$ reflects the strength of feature selection in sensitivity analysis. By adjusting the value of K, it is theoretically possible to achieve arbitrarily precise computational approximation with low power consumption simultaneously.

## 7 EXPERIMENTS

In this section, we apply our proposed method to the CIFAR-10 and CIFAR-100 datasets using the TET code framework for testing. Specifically, we employ the ResNET-19 and Spikedriven Transformer networks as our experimental models. Our method relies on the output at time $t$ to compute gradients and identify sensitive neurons at time $t + 1$. Consequently, the entire network operation cannot be processed in parallel and must be executed step-by-step in a serial manner. To accommodate this requirement, we have restructured the original code architecture to support serial processing, replacing the previous parallel computation mode. This modification ensures the accurate implementation of our experiments, tailored to the specific needs of our approach.

Table 2: Comparison of spike number and simulation length T

| Datasets | Architecture | Methods | SpikeNumber$\times 10^5$(Acc) | | | |
| --- | --- | --- | --- | --- | --- | --- |
| | | | T=1 | T=2 | T=4 | T=6 |
| CIFAR10 | ResNet19 | TET Deng et al. (2022) | 2.56(92.34) | 4.78(94.16) | 8.68(94.44) | 12.23(94.50) |
| | | RMP Guo et al. (2023a) | 2.45(93.45) | 4.51(94.26) | 8.43(94.37) | 13.14(94.48) |
| | | MPBN Guo et al. (2023b) | 2.27(93.45) | 3.34(93.55) | 7.49(93.60) | 10.97(93.90) |
| | | TAB Jiang et al. | 2.16(93.42) | 4.44(94.52) | 7.13(94.53) | 11.96(94.55) |
| | | DDLA(Ours) | **2.56(92.34)** | **3.32(93.92)** | **6.32(94.32)** | **7.01(94.38)** |
| | Transformer-2-256 | Spike-Driven-Transformer Yao et al. (2024) | 0.90(90.72) | 1.79(93.51) | 3.38(94.23) | 5.12(94.32) |
| | | DDLA(Ours) | 0.90(90.72) | **1.65(93.24)** | **2.84(94.02)** | **3.28(94.07)** |
| CIFAR100 | ResNet19 | TET Deng et al. (2022) | 2.47(69.44) | 5.61(71.92) | 10.46(74.47) | 15.94(74.72) |
| | | RMP Guo et al. (2023a) | 2.92(71.84) | 5.22(73.34) | 10.04(73.71) | 14.69(73.81) |
| | | MPBN Guo et al. (2023b) | 2.75(70.84) | 4.06(71.28) | 8.74(72.56) | 15.12(73.11) |
| | | TAB Jiang et al. | 2.66(70.89) | 5.17(71.39) | 9.23(73.00) | 14.98(73.80) |
| | | DDLA(Ours) | **2.47(69.44)** | **4.50(71.35)** | **6.02(73.65)** | **6.73(73.85)** |
| | Transformer-2-256 | Spike-Driven-Transformer Yao et al. (2024) | 0.93(67.35) | 1.83(68.70) | 3.55(69.25) | 5.15(70.68) |
| | | DDLA(Ours) | 0.93(67.35) | **1.69(68.50)** | **2.87(69.01)** | **2.94(70.59)** |

The number of spike in Table 2 represents the total number of pulses required to complete one forward inference per image on average in the corresponding dataset.According to our experimental results, we observed variations in the number of spikes across different computational methods when there were no constraints on the spike count of the neural networks. However, under the same network architecture, these variations generally remained within the same order of magnitude. Additionally, the relationship between the number of spikes and time steps is approximately proportional, indicating that the differences in spike counts across different time steps are not significant. Despite this, observations within individual time steps typically show fewer spikes in later steps.

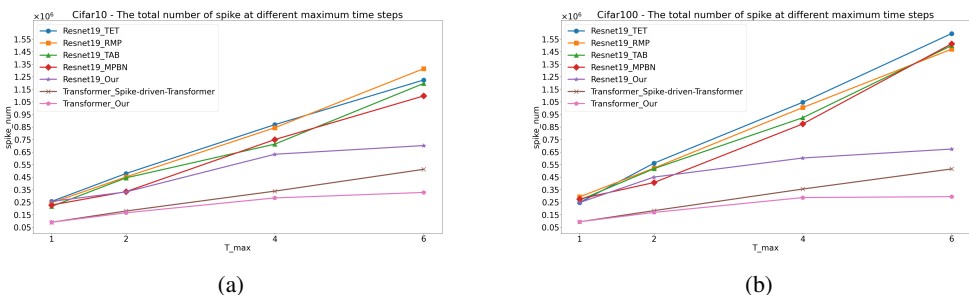

(a)                            (b)

Figure 4: This figure shows the average number of spike generated per image at different maximum time steps during the inference process on the (a) Cifar10 and (b) Cifar100 datasets.

During dataset comparisons, we particularly noted variations in spike distribution when the same network architecture processed different datasets. For instance, while handling the CIFAR-10 dataset, which has lower image complexity, the spike count was generally lower than when processing the more complex CIFAR-100 dataset.

Our experiments further revealed a close relationship between spike count and neural network accuracy. When facing the same dataset, setting a smaller maximum time step resulted in a higher spike count per time step. Conversely, increasing the maximum time step led to an overall increase in spike count but a decrease per time step, with only marginal improvements in recognition accuracy. This finding suggests that as the number of time steps increases, the informational value provided by the same amount of spikes gradually diminishes, indicating a decrease in the marginal utility of spikes.

After implementing DDLA, we observed not only a reduction in the overall number of spikes but also a significant decrease in the number of spikes per timestep, especially in cases with a larger maximum timestep. For instance, in the ResNet19 architecture on Cifar100, when T=6, the spike count was reduced by $57.8\%$, and according to Figure 5, the number of spikes at the sixth timestep

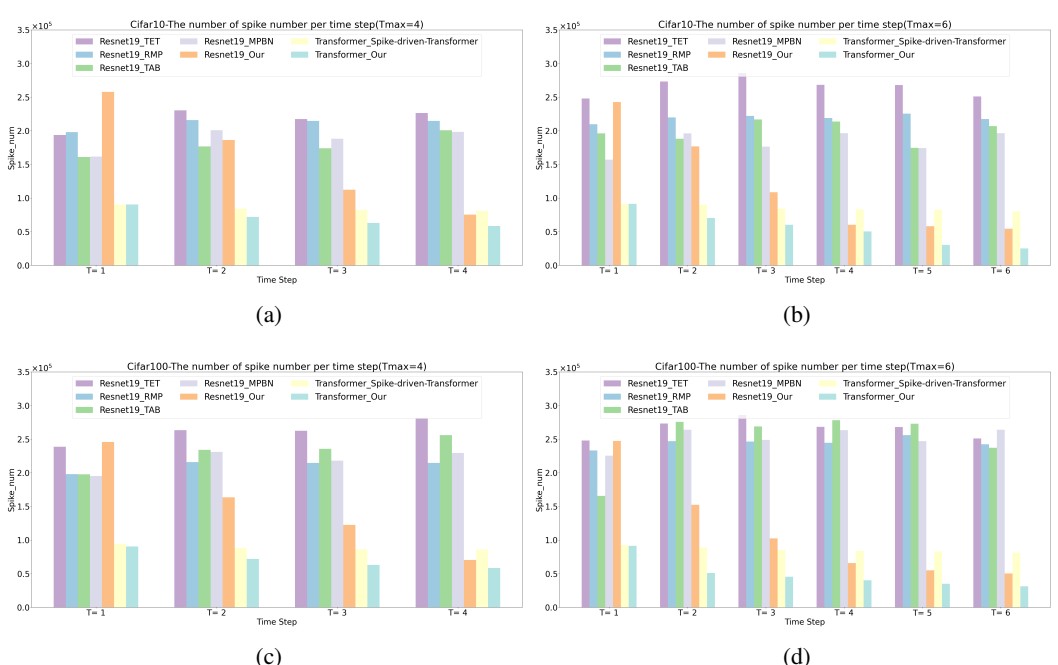

Figure 5: This figure shows the average number of spike generated by each image at different maximum time steps during the inference process on the Cifar10 and Cifar100 datasets.

decreased by approximately $80\%$. This marked reduction in spikes aligns with our expectations, as beyond a certain number of timesteps, additional spikes no longer provide extra information. This strategy not only optimizes the energy efficiency of SNNs but also opens possibilities for deploying efficient neural networks in resource-constrained environments. Our research provides new insights and empirical data on the energy efficiency and performance optimization of SNNs.

## 8 DISCUSSION

In this paper, we explore and analyze the phenomenon of feature superposition in SNNs, revealing the distribution patterns of features within the current SNNs computational frameworks. We propose a novel approach of shifting the network's focus from the feature domain to the temporal difference domain. This shift is predicated on the importance of feature dynamics over static features in SNNs, which not only reduces the number of spikes and conserves energy but also enhances the network's focus on changes between features. Concurrently, we introduce a preliminary method for constraining spike count in the temporal difference domain that aims to minimize spike emissions without compromising accuracy. It is important to note that our method is not the optimal solution but offers a new perspective and constitutes a modest step forward in this field. Our experiments were conducted on smaller datasets and network architectures, and the results demonstrate the efficacy of our approach. We encourage further research into the computational mechanisms of SNNs from the perspective of the temporal difference domain.

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

## A  APPENDIX

### A.1  PROOF OF THEOREM 1

We use the surrogate gradient and its primitive function method to approximate the output activation function, thus making the function have continuous derivatives of any order. Therefore, the original expression7 can be expanded into an integral order Taylor series around the approximation point. This Taylor series expansion is crucial for deriving both the exact and approximate formulas for error propagation. Let the spiking activation function be $\hat{f}_s$ and the surrogate gradient function be $\hat{f}'_s$, then during the training process, the function is expanded using the Taylor series:

$$
\begin{aligned}
O_l^t &= \hat{f}_s \left( f_l \left( O_{l-1}^{t-1} \right) + U_l^{t-1} + f_l(\Delta'^t_{l-1}) \right) \\
&= \hat{f}_s \left( f_l \left( O_{l-1}^{t-1} \right) + U_l^{t-1} \right) + \nabla \hat{f}_s^\top \left( f_l \left( \Delta'^t_{l-1} \right) \right) f_l \left( \Delta'^t_{l-1} \right) + R_1 \left( \Delta'^t_{l-1} \right)
\end{aligned}
\tag{16}
$$

where $\nabla \hat{f}_s^\top (X)$ is expressed by:

$$
\nabla \hat{f}_s^\top (X) = \left[ \frac{\partial \hat{f}_s}{\partial x_1}, \frac{\partial \hat{f}_s}{\partial x_2}, \cdots, \frac{\partial \hat{f}_s}{\partial x_n} \right] \Bigg|_X^\top = \left[ \hat{f}'_s (x_1), \hat{f}'_s (x_2) \cdots \hat{f}'_s (x_n) \right] \Big|_X^\top
$$

From the conclusion in Jie Xue & Jianghong Ma (2012), we can deduce that the error is related to the second central moment about the origin. That is to say, $Var = \sum_{i=1}^n \left( \frac{\partial f(\mathbf{x})}{\partial x_i} \right)^2 \sigma_i^2$. Therefore,

$$
\epsilon^* = \sum_{l=1}^L \sum_{i,j} \left( \frac{\partial \hat{f}_s(\Gamma_l)}{\partial \gamma_i} \cdot \frac{\partial \hat{f}_s(\Gamma_l)}{\partial \gamma_j} \right) \mathrm{Cov} \left( f_k \left( \Delta'^t_{l-1} \right)_i, f_k \left( \Delta'^t_{l-1} \right)_j \right)^2
$$

## A.2 PROOF OF THEOREM 2

From Theorem 1, we can obtain

$$\|\epsilon^*\|_2 = \sum_{l=1}^{L} \left\|\nabla f_s^T(\Gamma_l)\right\|_2^2 \left\|\mathrm{Cov}\left(f_k\left(\Delta_{l-1}'^t\right)\right)\right\|_2 \tag{17}$$

The norm of the covariance matrix is related to the norm of the differential quantity $\|\Delta_{l-1}'^t\|$. From the definition of the differential quantity, it follows that

$$\|\mathrm{Cov}(f_l\left(\Delta_l'^t\right))\|_2 = \frac{1}{N}\left\|f_1\left(\|\Delta_{l-1}'^t\right)^\top f_l\left(\|\Delta_{l-1}'^t\right)\right\|_2$$

$$= \frac{1}{k^2 N}\left\|f_l\left(\Delta_l^t\right)\right\|^2 = \frac{1}{k^2 N}\left\|\Delta_l^t\right\|^2\|W_l\|^2 \tag{18}$$

where

$$\|\Delta_l^t\| = \|s(t) \oplus s_l(t-1)\|_2 = \|s(t) - s_l(t-1)\|_2 \tag{19}$$

For neurons following dynamics in Eq. 1- 4 in the main text, Eq. 4 can be rewritten as:

$$u_1(t) = u_l(t-1) + W_l s_{l-1}(t) - r_l(t)s_l(t) \tag{20}$$

Noting that the equation can be accumulated from $t = 1 \ldots T$, we can obtain:

$$u_l(T) = \sum_{t=1}^{T} W_l s_{l-1}(t) - \sum_{t=1}^{T} r_l(t)s_l(t) \tag{21}$$

Define $h_l(t) = \sum_{i=1}^{t} s_l(i)$ as firing rate,

$$u_l(t) = W_1 h_{1-1}(t) - \sum_{i=1}^{t} r_l(i)s_l(i) \tag{22}$$

From the Assumption 1, the lower bound of the membrane potential can be obtained. If each norm of the membrane potential is less than $V_t h$, then the upper bound of the membrane potential norm is $\|u_l(t)\|_2 \leq \sqrt{mn}V_{th}$. Therefore,

$$\varphi \leq \|u_l(t)\|_2 = \left\|W_1 h_{1-1}(t) - \sum_{i=1}^{t} r_l(i)s_l(i)\right\|_2 \leq \sqrt{N}V_{th} \tag{23}$$

Time step integration:

$$\varphi \leq (W_l h_{l-1}(T) - V_{th}h_l(T))T \leq \sqrt{mn}V_{th} \tag{24}$$

From this, the upper and lower bounds of the neuron firing rate can be obtained.

$$\frac{W_l h_{l-1}(t) - \sqrt{mn}V_{th}}{tV_{th}} \leq h_l(t) \leq \frac{W_l h_{l-1}(t) - \varphi}{tV_{th}} \tag{25}$$

From Eq. 26, the upper bound of the norm of the differential quantity can be obtained, and the minimum sparsity of the differential quantity can also be determined.

$$\|\Delta_l^t\| \leq 2\|h_l(t) - h_l(t-1)\|_2 \leq 2\frac{\sqrt{NV_{th} - \varphi}}{TV_{th}} \tag{26}$$

By stacking Eq. 15 and Eq. 23, we have

$$\|\mathrm{Cov}(f_l\left(\Delta_l'^t\right))\|_2 \leq \frac{4\left(\sqrt{N}V_{th} - \varphi\right)^2}{k^2 NT^2 V_{th}^2}\|W_l\|_2^2\left\|\nabla f_s^T(\Gamma_l)\right\|_2^2 \tag{27}$$

Combining Eq. 15 and Eq. 27, we obtain:

$$\|\epsilon^*\|_2 \leq \sum_{l=1}^{L} \frac{4\left(\sqrt{N}V_{th} - \varphi\right)^2}{k^2 N T^2 V_{th}^2} \|W_l\|_2^2 \left\|\nabla f_s^T(\Gamma_l)\right\|_2^2 \tag{28}$$

## A.3 ALGORITHM

---

**Algorithm 1** Overall training algorithm

---

1: **Input:** input $(X)$, output $(Y)$, weights $(W)$, threshold voltage $(V)$, layers $(L)$
2: # Forward propagation
3: **for** $t = 1$ to $T$ **do**
4:     $O_0^t \leftarrow$ Poisson Encoder $(X)$
5:     **for** $l = 1$ to $L - 1$ **do**
6:         **if** $t == 1$ **then**
7:             $F_{l-1}^t = X(t)$
8:         **end if**
9:         $\mathrm{grad\_}F_l = \left|\frac{\partial F_{l+1}^t}{\partial F_l^t}\right|$
10:         sorted_indices = Sort_Indices_By_Value($\mathrm{grad\_}F_l$)
11:         $F'_l[\text{sorted\_indices}[i]] = F_l[\text{sorted\_indices}[i]]$
12:         $\mathcal{L}_{\text{sensi}} = \text{MSE}\left(F'_l - F_l\right)$
13:     **end for**
14: **end for**
15: # calculate total loss
16: $\mathcal{L}_{\text{total}} = \lambda \mathcal{L}_{\text{model}} + (1 - \lambda) \sum_{l=1}^{n} \mathcal{L}_{\text{sensi}}^{(l)}(t, s)$
17: # Backward Propagation
18: **for** $t = 1$ to $T$ **do**
19:     **for** $l = 1$ to $L - 1$ **do**
20:         $\frac{dL_{\text{total}}}{dU_l^t} = \frac{dL_{\text{total}}}{dF_l^t} \frac{dF_l^l}{dU_l^t}$
21:     **end for**
22: **end for**

---

