# OpenReview forum: "Feature Overlapping: The Computational Redundancy Caused by Repeated Features Across Different Time Steps in SNNs"
_ICLR.cc/2025/Conference — ICLR 2025 Conference Withdrawn Submission_

### Official Review · Reviewer_wXoo · 2024-10-19

**Soundness:** 2
**Presentation:** 2
**Contribution:** 2
**Rating:** 3
**Confidence:** 4

**Summary:**

This paper points out the feature overlapping phenomenon in SNNs, proposes a temporal differential decoupling (TDD) to separate static and dynamic features to reduce the computational overhead, and further reduces the overhead by selecting only sensitive features based on the gradient sensitivity criterion (GSC).

**Strengths:**

1. This paper points out the feature overlapping phenomenon in SNNs.
2. The authors propose TDD to separate static and dynamic features, and only compute dynamic features to reduce overhead.
3. The authors propose a GSC that further reduces the overhead by selecting only the sensitive features and setting the other features to zero.
4. The authors conducted experiments on CIFAR10/100 to demonstrate the advantages of spike count reduction.

**Weaknesses:**

1. The feature overlapping phenomenon seems to occur only on static images, and whether it exists on neuromorphic data, which is more suitable for SNNs, is unknown. Since neuromorphic data is characterized by pronounced temporal patterns, this phenomenon may not be significant for neuromorphic data. The methods proposed in this paper have strong limitations.

2. The authors have only conducted preliminary experiments on CIFAR10/100, and it remains unknown whether the proposed method is feasible for large datasets such as ImageNet.

3. For CIFAR10/100, established SNNs have been able to achieve great results at very low time steps, e.g. 1 or 2 time steps. Larger time steps do not lead to large performance gains, and the significance of this work is further suppressed by the fact that feature overlap is even less pronounced at small time steps.

4. The presentation of this paper is confusing and the author is advised to improve the presentation. For example, the authors do not explicitly describe whether the TDD separation of static and dynamic features is followed by setting the static features to 0 or how they are handled (I inferred from the GSC setting the output of insensitive neurons to 0 that this is also the case when the TDD is separated). In another example, the authors placed the core process of the proposed methods in the algorithm in the appendix without a detailed explanation, again confusing. In particular, line 716 states that F^t_{l-1} = X(t) when t == 1. This should only be true for the input layer, and I hope the authors check this.

5. Still about the presentation of this paper. The authors claim to have experimented with event-based datasets in line 91 of the paper, but no event dataset is seen in the paper.

I suggest that the authors refine the presentation of this paper and the experiments to support the significance and effectiveness of the methods.

**Questions:**

Please see weaknesses.

---

### Official Review · Reviewer_FgPF · 2024-10-25

**Soundness:** 3
**Presentation:** 2
**Contribution:** 2
**Rating:** 3
**Confidence:** 4

**Summary:**

The authors introduce a new technique for reducing the spike counts in SNNs trained over short time spans for image classification while retaining classification accuracy. They do this by noting that network features are temporally correlated in time and make use of this observation in deriving a new algorithm which they experimentally validate on CIFAR10 and CIFAR100.

**Strengths:**

The proposed method lowers spike counts on image classification tasks compared to other SNN methods using larger time steps (T=6) while maintaining similar classification performance. The authors compare their method to several prior SNN models and the paper is generally well written - the authors made an effort to make their mathematical equations more presentable using appropriate color-coding (Table 1).

**Weaknesses:**

I have a few reservations regarding this work:
1. **Unclear how computational resource consumption is reduced**. I understand your method reduces the spike count for longer time steps (although arguably simulating SNNs for T=6 steps is not that long). It is unclear how this property relates to reducing the computational cost per time step. Is this for training, inference or both? I may be missing something, but I imagine all training is still performed sequentially on a GPU, so I don't see how training would be faster or require less memory.

2. **Unclear if your model is better**. In your comparisons in Table 2 other methods get a similar or higher accuracy using less time steps than your method while using fewer spikes. For example, on CIFAR10, TAB Jiang et al. is 4.44(94.52) at T=2 vs your method 7.01(94.38) at T=6. I am not convinced your method is better in this regard. Perhaps your method performs better on more challenging datasets that require longer time steps?

3. **Missing analysis and controls**. Relating to point 1, I repeatedly read in the paper that your method is computationally more efficient, but on which metrics? If it uses less memory, then show this and compare to related work [1]. If it trains faster, then show this and compare to related work [2]. I would also urge the authors to contrast their method to SNNs in which neurons spike at most once, as these networks have been shown to perform relatively well on various datasets using minimal spikes [3]. It would also be insightful to compare to a SNN trained using surrogate gradients with an activity penalty.

4. **Questionable biological relevance**. The authors state that SNNs are similar to biology in their introductory paragraph. But I question the biological relevance of their model as 1. it seems that only integrate-and-fire neurons and not leaky integrate-and-fire neurons were explored (would your method work with LIF?) and 2. their model was only trained for 6-time steps (which is very short when modelling the biology). The divergence from the biology is okay, but I would perhaps mention that this work is less applicable to modelling real neurons in the discussion.

[1] Perez-Nieves, N. and Goodman, D., 2021. Sparse spiking gradient descent. Advances in Neural Information Processing Systems, 34, pp.11795-11808.

[2] Taylor, L., King, A. and Harper, N.S., 2024. Addressing the speed-accuracy simulation trade-off for adaptive spiking neurons. Advances in Neural Information Processing Systems, 36.

[3] Hwang, S. and Kung, J., 2024. One-Spike SNN: Single-Spike Phase Coding with Base Manipulation for ANN-to-SNN Conversion Loss Minimization. IEEE Transactions on Emerging Topics in Computing.

**Questions:**

### Questions
- What is a third-generation neural network? (line 39)
- On line 91 you state that you used event-based datasets but I could not find any?
- Do you provide any implementation details (e.g. batch size, learning rate, epochs / or how the images were fed as input to the spiking network)?
- Is your code available?

### Additional feedback to improve your paper
- As per the ICLR guidelines, when the authors or the publication are not included in the sentence, the citation should be in parenthesis. For example, see line 38 - you should use \citep.
- Li et al. reference twice on line 46.
- I would suggest denoting the Heaviside step function (Eq. 2) as H to be consistent with the literature (minor point).
- Add color bars to Figure 1
- Typo on line 375 where it looks like you are rendering f_( instead of f(
- Line 450 there should be a whitespace before "According"
- I would make the labels bigger in Figures 4 and 5 to make the text more readable to the reader
- I would suggest adding error bars to Figure 5 (you should have the data to do so).
- It would perhaps also be interesting to quantify the correlation of features in time

---

### Official Review · Reviewer_GdYK · 2024-10-29

**Soundness:** 2
**Presentation:** 2
**Contribution:** 2
**Rating:** 3
**Confidence:** 4

**Summary:**

This paper aims to reduce the number of spikes required during inference to enhance the energy efficiency of spiking neural networks (SNNs). To achieve this, the authors first identify the phenomenon of feature overlapping, where temporal feature components are often redundantly calculated. They then propose the Gradient Sensitivity Criterion (GSC) to identify important spikes. These contributions form the Differential Domain Low-Sparsity Approximation (DDLA) algorithm. The proposed method is evaluated on the CIFAR-10 and CIFAR-100 datasets, demonstrating a significant reduction in the number of spikes with only a minimal loss in accuracy.

**Strengths:**

1. This paper focuses on a critical problem in energy-efficient inference for SNNs: reducing the number of spikes without compromising accuracy.
2. The proposed method demonstrates strong performance on the evaluated datasets.

**Weaknesses:**

1. The observed feature overlapping is an inherent characteristic of the LIF model. The feature presented in Table 1 refers to the membrane potential $u$. Since $u[t]$ is derived exclusively from $u[t-1]$, this temporal dependency implies that the membrane potential naturally encapsulates historical information. Additionally, the statement in Line 264 that “temporal feature components are often repeatedly calculated across multiple time steps” is misleading, as the calculation of $u[t]$ relies solely on $u[t-1]$ without necessitating the recomputation of $\\{u[t^{\prime}]\\}_{t^{\prime}=1}^{t-2}$.

2. The DDLA Algorithm is not explained clearly. The notations used in Algorithm 1 are not adequately defined, making it difficult to understand the algorithm's implementation. Furthermore, the application of the gradient sensitivity criterion within the DDLA algorithm to reduce spikes is not sufficiently detailed.

3. The performance of the baseline without the DDLA method is not presented, making it unclear the actual spike reduction and accuracy loss attributable to the DDLA method.

4. The contribution section states that the proposed method is evaluated on event-based datasets; however, I could not find any supporting evidence for this claim.

5.  The absence of experiments on large-scale datasets, such as ImageNet, raises questions about the scalability of the proposed method.

**Questions:**

1. Can the gradient sensitivity criterion effectively distinguish between dynamic and static features as defined in the paper?

2. What is the performance of the proposed DDLA  method on more widely used datasets beyond CIFAR-10 and CIFAR-100?

3. Is the DDLA method superior to a simple baseline that trains SNNs using a spike count regularization term?

---

### Official Review · Reviewer_ti4h · 2024-11-02

**Soundness:** 3
**Presentation:** 3
**Contribution:** 2
**Rating:** 6
**Confidence:** 4

**Summary:**

This paper is the first to identify the "Feature Overlapping" phenomenon in the computational process of Spiking Neural Networks (SNNs). The authors propose the Temporal Differential Decoupling (TDD) method, which separates dynamic and static features to reduce redundant computations. By utilizing the Gradient Sensitivity Criterion (GSC) and the Differential Domain Low-Sparsity Approximation (DDLA) algorithm, the approach effectively minimizes computational resource consumption. Experimental results show a significant reduction in the number of spikes and inference costs.

**Strengths:**

The paper paper analyzes feature extraction in Spiking Neural Networks (SNNs) from a novel perspective.
##
The paper paper introduces  new interesting algorithms aimed at reducing the power consumption of SNNs.

**Weaknesses:**

The paper lacks a comparison with other methods for reducing SNN power consumption. I suggest that the authors include such an analysis to provide a clearer understanding of the advantages and limitations of the proposed approach.
##
Since the experiments are based on static datasets, it is unclear whether the feature redundancy phenomenon can also be observed in time series tasks.

**Questions:**

Static datasets lack true dynamics. How would the results change if the images were input directly without Poisson processing?
##
In Table 2, why is it possible to perform temporal feature decoupling even when T=1?
##
The font size in the figures could be appropriately increased.

---

### Note · Authors · 2024-11-13

I have read and agree with the venue's withdrawal policy on behalf of myself and my co-authors.